# Investigation of Therapeutic Response Markers for Acupuncture in Parkinson’s Disease: An Exploratory Pilot Study

**DOI:** 10.3390/diagnostics11091697

**Published:** 2021-09-17

**Authors:** Sangmin Park, Aeyung Kim, Gunhyuk Park, Ojin Kwon, Sangsoo Park, Horyong Yoo, Junghee Jang

**Affiliations:** 1KM Data Division, Korea Institute of Oriental Medicine, 1672 Yuseongdae-ro, Yuseong-gu, Daejeon 34054, Korea; kiro123@kiom.re.kr; 2KM Application Center, Korea Institute of Oriental Medicine, 70 Cheomdan-ro, Dong-gu, Daegu 41062, Korea; aykim71@kiom.re.kr; 3Herbal Medicine Resources Research Center, Korea Institute of Oriental Medicine, 111 Geonjae-ro, Naju-si 58245, Korea; gpark@kiom.re.kr; 4KM Science Research Division, Korea Institute of Oriental Medicine, 1672 Yuseongdae-ro, Yuseong-gu, Daejeon 34054, Korea; cheda1334@kiom.re.kr; 5Clinical Trial Center, Daejeon Korean Medicine Hospital, 75, Daedeok-Daero 176 Beon-gil, Seo-gu, Daejeon 35235, Korea; pssna007@hanmail.net

**Keywords:** Parkinson’s disease, acupuncture, transcriptome, SYN1, ANKRD22

## Abstract

In this preliminary pilot study, we investigated the specific genes implicated in the therapeutic response to acupuncture in patients with Parkinson’s disease (PD). Transcriptome alterations following acupuncture in blood samples collected during our previous clinical trial were analyzed along with the clinical data of six patients with PD, of which a representative patient was selected for transcriptomic analysis following acupuncture. We also examined the changes in the expression of PD biomarker genes known to be dysregulated in both the brain and blood of patients with PD. We validated these gene expression changes using quantitative real-time polymerase chain reaction (qPCR) in the blood of the remaining five patients with PD who received acupuncture treatment. Following acupuncture treatment, the transcriptomic alterations in the representative patient were similar to those induced by dopaminergic therapy. Among the PD biomarkers, ankyrin repeat domain 22 (ANKRD22), upregulated following dopaminergic therapy, and synapsin 1 (SYN1), a common gene marker for synaptic dysfunction in PD, were upregulated following acupuncture. These alterations correlated with changes in gait parameters in patients with PD. Our data suggest ANKRD22 and SYN1 as potential biomarkers to predict/monitor therapeutic responses to acupuncture in patients with PD, especially in those with gait disturbance. Further research is needed to confirm these findings in a large sample of patients with PD.

## 1. Introduction

Parkinson’s disease (PD), a neurological disorder, is characterized by multiple symptoms caused by circuit dysfunctions attributed to the depletion of striatal dopamine [1]. Acupuncture has been widely used to ameliorate both motor and non-motor symptoms of PD [2,3,4,5,6]. Previous clinical trials have reported that acupuncture improved the gait parameters, quality of life, and Unified Parkinson Disease Rating Scale II and III scores of patients with PD [2,3,4,5]. Acupuncture stimulation has also been reported to enhance the survival of dopaminergic neurons, attenuate the increase in alpha-synuclein (α-syn) in the substantia nigra, and improve motor function in a 1-methyl-4-phenyl-1,2,3,6-tetrahydropyridine (MPTP) parkinsonism mouse model [7,8].

The molecular mechanisms underlying PD pathogenesis remain poorly understood and specific biomarkers for earlier intervention to improve PD prognosis and therapeutic response remain lacking [9]. The advent of high-throughput sequencing technology enables genome-wide analysis that can investigate the transcriptome, which is the set of all RNA transcripts [10]. Therefore, several transcriptomic studies have employed genome-wide expression profiling using RNA sequencing (RNA-Seq) data to elucidate the pathogenic mechanisms of PD and to identify potential genetic biomarkers by examining changes in expression levels induced by disease state or treatment [9]. Although using peripheral blood transcriptome analysis to identify PD biomarkers in the blood has been limited to the central nervous system (CNS) pathology of PD, certain gene alterations are consistent in both the brain and blood; for example, protein and mRNA levels of p11, also known as S100 calcium-binding protein A10, are decreased in both the postmortem brains and the blood cells of patients with PD [11]. A recent analysis of the brains of living patients with PD also found similar genetic biomarkers in the blood cells [12], including *ANKRD22*, *IL1R2*, and *MARCH1*. This consistency may be attributed to central and peripheral PD-induced pathological alterations [12]. Furthermore, since the striatal blood-brain barrier is impaired in PD [13], changes in blood transcriptomes may reflect the CNS pathology. Based on these findings, we hypothesized that transcriptomic changes in the blood following therapeutic interventions could perhaps reflect neural changes in patients with PD.

This study aimed to demonstrate the feasibility of using RNA-Seq of blood cells to predict the therapeutic response of patients with PD to acupuncture. Our previous clinical trial had reported that the hypsometric gait of patients with PD improved after acupuncture in terms of a decreased cadence and increased stride time and length [14]. In addition, the hemodynamic response in the prefrontal cortex (PFC) and supplementary motor area (SMA) significantly increased following acupuncture. We hypothesized that transcriptomic alterations associated with improved gait disturbance and rearranged cerebral cortex activity could also be identified in the blood of patients with PD. The newly discovered acupuncture-induced transcriptomic alterations provide novel insights into the molecular mechanisms underlying the effects of acupuncture and the use of blood biomarkers to predict (or monitor) therapeutic responses in patients with PD.

## 2. Materials and Methods

This study was approved by the Clinical Trial Center of Daejeon Korean Medicine Hospital, Daejeon University repository (IRB number DJDSKH-17-BM-20) and registered at the Clinical Research Information Service (registry number KCT0002603). A total of six participants were enrolled in this study after providing signed written informed consent to participate.

### 2.1. Acquisition of Clinical Data

Clinical data regarding the gait of patients with PD before (V1) and after eight sessions (V8) of acupuncture were obtained from a previous clinical trial that assessed the following gait parameters using the GAITrite system (CIR Systems Inc) [14]: velocity (cm/s), cadence (steps/min), stride time (s), stride length (cm), single support (s), double support (s), swing time (s), and swing % cycle. Moreover, changes in oxyhemoglobin levels were used to assess the hemodynamic response in the cerebral cortex using functional near-infrared spectroscopy (fNIRS) before and after acupuncture [14]. The blood samples of patients with PD were collected before (V1) and after five (V5) as well as eight sessions (V8) of acupuncture. Detailed information about the study design, inclusion criteria, interventions, and outcome measurements can be found in the clinical protocol study [15].

### 2.2. Quality Test of Blood Samples

Total RNA was extracted from all the blood samples using the PAXgene Blood RNA Kit (Qiagen, Hilden, Germany); the quantity and quality of RNA were assessed using a NanoDrop ND-1000 Spectrophotometer (NanoDrop Technologies Inc., Rockland, DE, USA). RNA samples for subsequent analysis were selected based on optical densities of 260/280 and 260/230 and the amount of RNA. 

### 2.3. Acquisition of RNA Sequencing (RNA-Seq) Data

The PAXgene Blood miRNA Kit (Qiagen) was used for total RNA isolation from the whole blood samples of a representative patient before and after V5 and V8. RNA quality was assessed using an Agilent 2100 bioanalyzer and the RNA 6000 Nano Chip (Agilent Technologies, Amstelveen, The Netherlands), and RNA quantification was performed using an ND 2000 spectrophotometer (Thermo Inc., Lewes, DE, USA). RNA-Seq was performed commercially by Ebiogen Inc. (Seoul, Korea). Library construction for control and test RNAs was performed using the QuantSeq 3′mRNA-Seq Library Prep Kit (Lexogen Inc., Wien, Austria) according to the manufacturer’s instructions. QuantSeq 3′mRNA-Seq reads were aligned using Bowtie2 [16]. Bowtie2 indices for alignment to the genome and transcriptome were generated from either the genome assembly or the representative transcript sequences. The alignment file was used for assembling transcripts, estimating their abundances, and detecting differential gene expression.

### 2.4. Transcriptomic Analysis

A fold change describes the ratio of two values, measuring how much a quantity changes between an original and a subsequent measurement. Compared to the baseline levels at V1, fold changes after V5 or V8 were calculated for each gene based on count-per-million values using *edgeR* package in R [17] after filtering out low count genes (raw count < 3 in any sample). Differentially expressed gene (DEG) analysis for the treated (V5 and V8) states versus the untreated (V1) state was also conducted using the *edgeR* package in R.

To compare the therapeutic effects, the gene sets with significant differential expression in patients with idiopathic PD before and after dopaminergic therapy were obtained (69 upregulated and 21 downregulated genes) [18]. Single-sample gene set enrichment analysis (ssGSEA) was performed using *GSVA* package in R [19]. The Gene Ontology (GO) biological process (BP) gene set was downloaded from MsigDB v.7.1 (https://www.gsea-msigdb.org/gsea/msigdb).

### 2.5. Comparison of PD Biomarkers

Changes in biomarkers that are common in the brain tissue and blood of patients with PD were examined. The gene sets showing consistently altered expression in the frontal lobe and blood of living patients with PD were obtained, comprising two upregulated (*BTNL9* and *STOX1*) and four downregulated genes (*ANKRD22*, *IL1R2*, *MARCH1*, and *OLFML2B*) [12]. We also integrated the data from multiple PD biomarker studies that focused on dysfunctions in neurotransmission. Analysis of postmortem substantia nigra and striatum tissues revealed altered expression genes, which were also validated using qPCR [20]. Analysis of postmortem dopaminergic neurons has also shown deregulated genes for neurotransmission [21]. Network analysis combining multiple datasets from several brain regions, including the cortex, showed the five most highly connected hub nodes (*MAPK8*, *RAB3A*, *STXBP1*, *SYN1*, and *VAMP2*) in a reconstructed PD disease direct interaction network [22]. Consistently, analysis of the blood of patients with PD highlighted four genes (*SYN1*, *GRIN1*, *GRIN2D*, and *DLGAP3*) involved in the neuronal system pathway [23]. The fold changes for the above genes were analyzed.

### 2.6. Quantitative Real-Time Polymerase Chain Reaction (qPCR)

Total RNA (500 ng) isolated from whole blood samples was reversed transcribed into cDNA using the Superscript III First-Strand Synthesis System (Invitrogen, Carlsbad, CA, USA). After diluting cDNA to a constant concentration, qPCR was performed using PowerSYBR Green PCR Master Mix (Thermo Fisher Scientific, Waltham, MA, USA). The qPCR reactions were set up in triplicate with a total volume of 20 µL: 2 µL of cDNA template, 1 µL of each primer (5 pmol), 10 µL of Power SYBR Green PCR Master Mix (2×), and 6 µL of nuclease-free water. qPCR was performed using QuantStudio 6 Flex Real-Time PCR System (Applied Biosystems, Foster City, CA, USA) and reaction condition was as follows: 40 cycles of DNA denaturation at 95 °C for 15 s and annealing/extension at 60 °C for 1 min. The average cycle threshold (Ct) value of the biomarker genes was calculated post normalization to that of the GAPDH gene, and then the fold increase or decrease in gene expression compared to V1 sample was determined [24]. Gene-specific primers were as follows: *SYN1* (5′-CATCGATGCCAAATATGACG-3′, 5′-AGCATCGCAGAGCCAGTATT-3′); *ANKRD22* (5′-GACCCCACAATAAAGAATAAGC-3′, 5′-CCCACAGACCAAAAGTCTAAAA-3′); *IL1R2* (5′-TGTGCTGGCCCCACTTTC-3′, 5′-GCACAGTCAGACCATCTGCTTT-3′); *MARCH1* (5′-GCCTCACAAACCTCCACATT-3′, 5′-TCTGCAGATGTCCTGAGTGG-3′); and *GAPDH* (5′-CCTCAACGACCACTTTGTCA-3′, 5′-TCTTCCTCTTGTGCTCTTGC-3′). 

### 2.7. Statistical Analysis

The log2 (fold change) values from the RNA-Seq data were statistically analyzed using a two-tailed Student’s *t*-test. The expression levels measured using qPCR were statistically analyzed using a two-tailed paired *t*-test. After the normality test, Pearson’s correlation analysis was performed between gene expression levels and gait parameters.

## 3. Results

### 3.1. Selection of a Representative Patient with PD Based on Clinical Outcome Changes following Acupuncture

Using blood samples from patients with PD who underwent acupuncture treatment in our previous clinical study [14], we determined the changes in the molecular signature. Due to prolonged storage of the blood samples, some samples were excluded from subsequent analyses (Appendix A). Out of the 39 blood samples analyzed, RNA was undetected in only three samples, whereas nine failed the quality test since either the amount of RNA was insufficient or the optical density was outside the manufacturer’s normal range. Six patients whose blood samples passed the quality test at three timepoints (V1, V5, and V8) were selected for further analysis. The individual baseline data on demographics and gait patterns of the six patients are summarized in Table 1. 

To select a representative patient for accurately reflecting the therapeutic response to acupuncture in PD, we reanalyzed the individuals’ clinical information obtained from the previous study [14] (Appendix A). The patient with the greatest changes in cadence and stride length following acupuncture was selected as the representative patient (Figure 1A), who also showed increased oxyhemoglobin levels in the PFC and SMA (Figure 1B). We analyzed the blood transcriptome of the representative patient and experimentally validated our findings in the remaining five patients (Appendix A).

### 3.2. Transcriptomic Alterations following Acupuncture

We investigated the transcriptomic alterations in the blood of the representative patient using RNA-Seq data from samples obtained at V1, V5, and V8. Owing to the small sample size, our data are limited to statistical analysis of differential gene expression (Appendix A). In an alternative method, we used the gene sets that are differentially expressed following dopaminergic therapy in patients with PD [18] to verify that the transcriptomic alterations of the representative patient represent the therapeutic effect in PD. The 38 upregulated or six downregulated genes after dopaminergic therapy were referred to as UD (upregulated genes by dopaminergic therapy) or DD (downregulated genes by dopaminergic therapy) groups, respectively. Next, we analyzed the fold changes by acupuncture for each gene’s expression relative to the baseline level at V1 (Appendix A). To observe the distribution of expression changes by gene groups, we analyzed the data using a density plot, a smoothed version of histograms. The density plots of the fold-change levels showed increased and decreased distributions for the UD and DD groups, respectively, compared to global deviations in all genes at both V5 and V8 (Figure 2A). Compared to the genes in the DD group, the expression of genes in the UD group increased after both V5 (*p* = 0.0004) and V8 (*p* = 0.0582) (Figure 2B). These results suggested that acupuncture treatment induced transcriptomic alterations in the representative patient’s blood in a manner similar to dopaminergic therapy. In the UD group, *ANKRD22* showed the maximum average increase in expression at V5 and V8 (Appendix A).

To determine the effects of acupuncture on functional pathways in the representative patient, we performed pathway enrichment analysis using ssGSEA and obtained the degrees of change in enrichment scores for each functional term in the GO BP gene set after acupuncture treatment at V5 and V8 (Appendix A). The results showed that genes upregulated by acupuncture treatment were enriched in pathways involved in synaptogenesis, neurotransmission, and synaptic vesicle dynamics (Figure 2C). The pathway with the highest enrichment scores was involved in the positive regulation of inhibitory postsynaptic potential (IPSP). 

### 3.3. Expression Changes in Biomarkers in the Blood and Brain of Patients with PD following Acupuncture 

Among the previously reported PD biomarkers that were replicated in independent studies, we identified genes whose expression changes correlated with the therapeutic response to acupuncture. Blood transcriptome analysis is insufficient to investigate the pathologies of brain-associated diseases, such as PD; thus, to accurately evaluate expression changes in the brain using gene expression changes in the blood of patients with PD, we selected biomarkers that are common between the brain and blood of patients with PD. A recent study on the living brains of patients with PD [12] revealed DEGs consistent with previously reported biomarkers in the blood of patients with PD [25]. The expression levels of three genes, namely *ANKRD21*, *IL1R2*, and *MARCH1*, are lower in both the brain and the blood of patients with PD than in those of healthy controls [12]. As shown in Figure 3A, the expression of these genes increased following acupuncture treatment, which suggests that acupuncture may ameliorate the downregulation of these genes in both the brain and blood of patients with PD. Intriguingly, *ANKRD22*, which is upregulated after dopaminergic therapy in PD [18], was also upregulated after acupuncture treatment, highlighting the potential of *ANKRD22* as a response biomarker to predict/monitor therapeutic effects in patients with PD.

As another approach for examining the common genetic biomarkers in both the brain and blood of patients with PD, we integrated biomarkers reported from postmortem brain studies on neurons [21], brain tissues [20], network analysis [22], and blood [19] from patients with PD (Appendix A). *SYN1* was the common gene among all biomarkers associated with synaptic dysfunction in PD (Figure 3B, left). Intriguingly, *SYN1* was also the most upregulated gene following acupuncture among these biomarkers (Figure 3B, right). These results suggest that *SYN1* is a key biomarker indicative of deregulated synaptic vesicle dynamics across all tissues of patients with PD and can be recovered by acupuncture treatment.

### 3.4. Validation of the Identified Acupuncture Therapy Response Markers

The blood samples of the remaining five patients were analyzed to validate the results of the transcriptome alterations in the representative patient. To confirm the changes in the expression of *SYN1*, *ANKRD22*, *IL1R2*, and *MARCH1* following acupuncture treatment, we analyzed the mRNA levels of these genes in the blood samples obtained from these remaining patients (Figure 4A). We found that the levels of *SYN1* (*p* = 0.0574 and 0.0054) and *ANKRD22* (*p* = 0.0910 and 0.0678) showed an increasing trend following acupuncture treatment (V5 and V8, respectively). Correlation analysis between the change in the gait parameters and gene expression showed that the expression changes in *SYN1* (*p* = 0.0318, *r* = 0.6761) and *ANKRD22* (*p* = 0.0143, *r* = 0.7404) were positively correlated with changes in stride length (Figure 4B), thereby suggesting that *SYN1* and *ANKRD22* are acupuncture therapy response biomarkers in the blood of patients with PD.

## 4. Discussion

This study used blood transcriptomes to identify therapeutic response biomarkers following time-dependent acupuncture (baseline, five, and eight acupuncture sessions) for patients with PD. To our knowledge, this is the first report on gene expression changes in patients with PD whose gait––as measured through the GAITrite system––improved following acupuncture.

The analysis of blood transcriptome data from the representative patient with PD whose hypometric gait significantly improved following acupuncture indicated that changes in gene expression and transcriptomes are similarly affected by acupuncture as by dopaminergic treatment. The pathway enrichment analysis revealed that the expression of genes that positively regulate IPSP dramatically increased following acupuncture. Conversely, since excessive synchronization within the basal ganglia–cortical loop may contribute to motor symptoms in PD [26,27], the enhancement of IPSPs with acupuncture may improve motor impairment in patients with PD by inhibiting neuronal action potentials and reducing excessive synchronization. However, whether an increase in IPSP in the blood would affect the brain regions remains to be determined. 

The expression levels of *ANKRD22*, *IL1R2*, and *MARCH1*, which are downregulated in both the living brain and blood of patients with PD [12], and *SYN1* increased following acupuncture in the representative patient. Moreover, qPCR validation to determine the changes in gene expression patterns in the remaining five patients with PD, who underwent acupuncture, confirmed the possibility of upregulation in *ANKRD22* and *SYN1* following acupuncture. The statistical significance of the change in the expression level of *SYN1* and *ANKRD22* was increased during eight acupuncture treatments (V8) compared to five times (V5), suggesting that these therapeutic response biomarkers may increase with cumulative sessions of acupuncture treatment.

Studies regarding the transcriptomic alterations of *ANKRD22* in patients with PD are limited and the mechanisms by which *ANKRD22* transcriptomic alterations contribute to neurodegenerative diseases are poorly understood. A previous study on the ankyrin repeat domain in PD reported that ankyrin repeats and the anti-tumorigenic BTB/POZ domain-containing protein-2 (BPOZ-2) inhibit α-syn aggregation in 1-methyl-4-phenylpyridinium-stimulated dopaminergic neurons in an MPTP mouse model of PD [28]. α-syn aggregation may influence the inhibition of proteasomal degradation due to the dysfunction of parkin, an E3 ubiquitin ligase [29], and BPOZ-2, an ankyrin-rich scaffold protein that is known to control the activity of E3 ubiquitin ligases. Parkin may interact with the cullin family of E3 ubiquitin ligases, the binding partners of BPOZ-2. Therefore, crosstalk between BPOZ-2 and parkin may occur via ubiquitin ligase. Another study reported *PLA2G6* (*PARK14*), an important paralog of *ANKRD22*, as a parkinsonism-associated gene, which its mutation influences the onset of neurodegenerative disorders, including early-onset parkinsonism [30]. The loss of *iPLA2-VIA* (*Drosophila* homolog of *PLA2G6*) impairs retromer function and causes lysosomal and neuronal dysfunction [30]. This is similar to the effect of α-syn overexpression and may be one of the common pathological mechanisms underlying PD [30]. Weber et al. [31] reported that the ankyrin repeat domain in *Drosophila* controls pre-synaptic localization and is essential for synaptic stability. While the ankyrin repeat domain may be associated with the pathological mechanisms of PD, the exact role of *ANKRD22* in PD pathogenesis remains to be established. 

SYN1 codes for synapsin I, which belongs to a family of neuronal phosphoproteins that are known to modulate neurotransmitter release at pre-synaptic terminals and are involved in neuronal development and synaptic plasticity [32,33]. Specifically, synapsins have been reported to modulate synaptic vesicle mobilization, docking, fusion, and recycling upon neuronal stimulation in pre-synaptic compartments [32]. Therefore, synapsins function to control the trafficking and availability of synaptic vesicles [32]. In addition, treatment with neurotransmitters, such as dopamine and noradrenaline, induces phosphorylation of synapsin proteins in the brain [32]. Furthermore, upregulation of SYN1 is expected to be correlated with enhancement of pre-synaptic vesicle availability and neurotransmitter release following acupuncture.

A previous study proposed that acupuncture could enhance motor function and dopamine availability by increasing dopamine release in the synaptic cleft and mitigating abnormal post-synaptic changes in the MPTP PD mouse model [34]. MPTP inactivation decreased dopamine levels and increased FosB expression regulated by dopamine- and cAMP-regulated phosphoprotein of 32 kDa (DARPP-32) signaling in the striatum. Changes in FosB expression were determined based on MPTP-induced post-synaptic changes. Acupuncture improved motor function in the MPTP PD model mice by reducing MPTP-induced DARPP-32 phosphorylation and FosB expression [34]. Consistent with previous findings, we found that the FosB levels in the blood were slightly downregulated following acupuncture in patients with PD (fold change = 0.886 at V8; Appendix A). Although FosB expression pattern in the brain remains poorly understood, it is possible that *SYN1*-induced dopamine availability enhancement in the presynaptic compartments and post-synaptic changes, including FosB expression, normalized the basal ganglia activity. This could be correlated with hypometric gait improvement and cerebral cortical rearrangement in patients with PD. 

This study has some limitations, one of which is the small sample size, hence we did not identify novel DEGs. Rather, we evaluated changes in the pattern of expression of previously identified DEGs based on conventional dopaminergic therapy or known PD biomarkers. Moreover, we experimentally validated our findings using expression changes in candidate genes in five patients. Nevertheless, further research is needed to confirm therapeutic response biomarkers to acupuncture in patients with PD. Another limitation is the use of blood samples, not brain samples, for transcriptomic analysis in PD. To overcome this limitation, we used PD biomarkers that have been previously shown to be altered in both the brain and blood. The direct effects of acupuncture on the brain of patients with PD require future validation in a larger patient population. Additionally, some gene expression changes following acupuncture tended to be opposite to those of the UD or DD group. This may imply that the mechanisms of dopaminergic therapy and acupuncture treatment are different. In addition to the GAITrite system used in this study, entropy methods of evaluating the gait based on complexity theory can provide more information for measuring the therapeutic effect in patients with PD [35].

## 5. Conclusions

*SYN1* and *ANKRD22* were upregulated in patients with PD who showed improved gait following acupuncture. The upregulation of *SYN1* may imply enhancement of pre-synaptic vesicle availability and neurotransmitter release, whereas *ANKRD22* alteration may affect α-syn aggregation. Collectively, our results suggested that *SYN1* and *ANKRD22* are potential biomarkers that can be used to predict/monitor acupuncture treatment responses in patients with PD. 

## Figures and Tables

**Figure 1 diagnostics-11-01697-f001:**
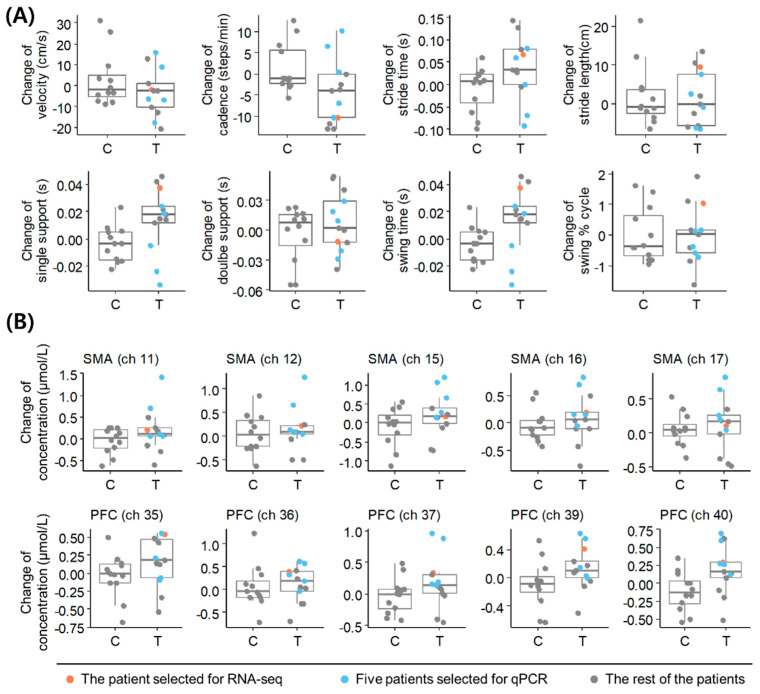
Clinical information for patients with PD. (**A**) Changes in gait parameters in the acupuncture treatment group (T) and control group (C) at V8 compared to V1. (**B**) Changes in oxyhemoglobin level in fNIRS channels in T and C at V8 compared to V1. Data of the representative patient selected for RNA−seq analysis and the other five patients selected for qPCR validation are highlighted with the indicated colors. SMA, supplementary motor cortex; PFC, dorsolateral prefrontal cortex; qPCR, quantitative real-time polymerase chain reaction; PD, Parkinson’s disease.

**Figure 2 diagnostics-11-01697-f002:**
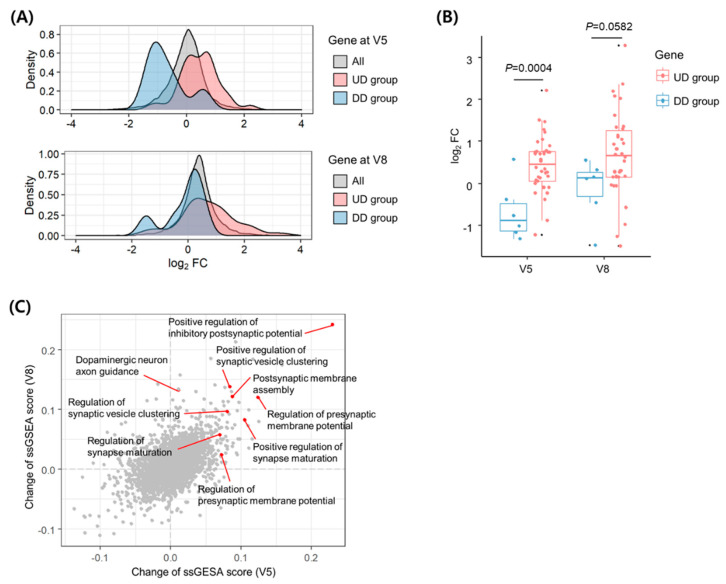
Transcriptomic alterations in the blood of the representative patient with PD. (**A**) Distributions of expression fold-changes for all genes, DD group, and UD group at V5 and V8 compared to V1. The density was defined as the proportion of genes with a specific fold change among all genes. (**B**) Comparison of the expression changes between genes in the UD and DD groups at V5 and V8 compared to V1. *P*−values were determined using the two-tailed Students’ *t*-test. (**C**) Changes in ssGSEA scores for the GO BP gene set at V5 and V8 compared to V1. Each point represents a changed set of scores in pathways of the gene set. The enriched pathways related to synapse transmission are indicated by red arrows. FC, fold change; ssGSEA, single-sample gene set enrichment analysis; UD, upregulated genes by dopaminergic therapy; DD, downregulated genes by dopaminergic therapy; PD, Parkinson’s disease.

**Figure 3 diagnostics-11-01697-f003:**
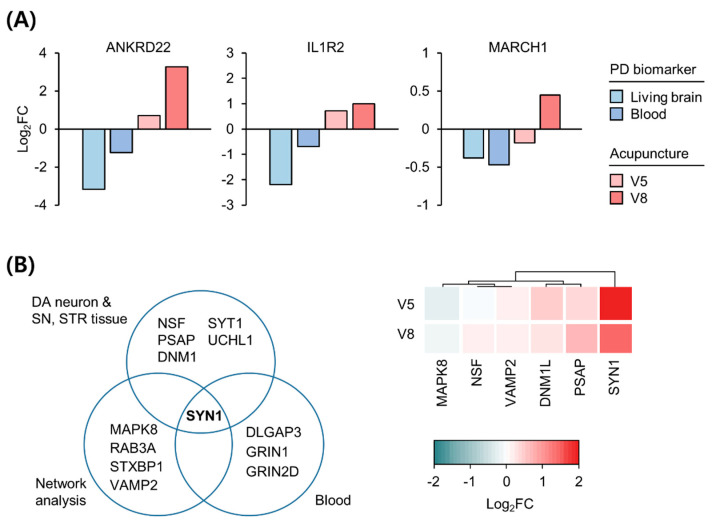
Expression changes in PD biomarkers following acupuncture treatment. (**A**) Three PD biomarkers consistently decreased in living brain and blood transcriptome versus healthy controls. Previously reported values of biomarker gene expression changes were compared with the changes in the representative patient’s blood at V5 and V8 compared to V1. (**B**) Suggested PD biomarkers based on postmortem brain tissue, blood, and network analysis (left). Expression changes in genes in the representative patient’s blood at V5 and V8 compared to V1 (right). Red and blue colors denote increased and decreased expressions, respectively. FC, fold change; PD, Parkinson’s disease.

**Figure 4 diagnostics-11-01697-f004:**
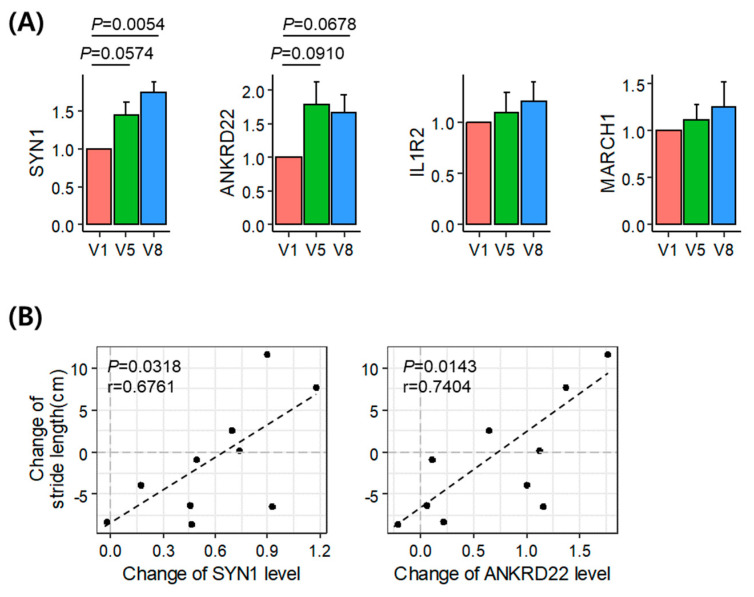
Experimental validation of the identified therapeutic marker genes. (**A**) Expression levels of the marker genes in the blood of five patients at V5 and V8 compared to V1. *P*-values were determined using the two-tailed paired *t*-test. (**B**) Correlation between changes in stride length and changes in SYN1 (left) and ANKRD22 (right) expression levels. The Pearson’s correlation coefficients (*r*) and *P*-values (*P*) indicate positive correlations. (It should be noted that results using Spearman’s rank-order correlation were less significant as *P*-value of 0.0667 and 0.0289 for SYN1 and ANKRD22, respectively).

**Table 1 diagnostics-11-01697-t001:** Individual baseline demographics and gait characteristics.

Case	Age	Sex	Age at Onset (Years)	Disease Duration (Years)	Hoehn and Yahr Scale	Velocity (cm/s)	Cadence (Steps/min)	Stride Time (s)	Stride Length (cm)	Single Support Time (s)	Double Support Time (s)
T02 *	69	M	64	5	3	155.433	135.567	0.885	138.270	0.360	0.162
T04	60	M	56	4	2	133.200	121.900	0.986	131.248	0.373	0.250
T09	75	M	73	2	2	103.500	104.567	1.143	119.055	0.441	0.261
T11	59	M	38	21	1	146.500	134.567	0.888	130.540	0.346	0.195
T12	70	M	63	7	1	102.800	118.000	1.018	104.502	0.381	0.262
T13	57	F	51	6	1	96.333	108.400	1.106	106.441	0.421	0.269

T, Treatment group with acupuncture; M, Male; F, Female; * The representative case patient for RNA-seq.

## Data Availability

The RNA-seq datasets generated for this study can be found in the Gene Expression Omnibus: accession number GSE178470 (https://www.ncbi.nlm.nih.gov/geo/query/acc.cgi?acc=GSE178470). All data needed to evaluate the conclusions of the study are present in the paper. The source document data used to support the findings of the present study have been deposited in the Clinical Trial Center of Daejeon Korean Medicine Hospital, Daejeon University repository (IRB number DJDSKH-17-BM-20).

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
