# Peer review of "Investigation of Therapeutic Response Markers for Acupuncture in Parkinson’s Disease: An Exploratory Pilot Study"

_diagnostics, 2021, doi:10.3390/diagnostics11091697_

Round 1

Reviewer 1 Report

This preliminary pilot study investigated the effect of acupuncture on transcriptome alterations and gene expression response in patients with PD and found that up-regulation of specific gene ANKRD22 and SYN1 and suggests these are potential biomarkers to predict/monitor therapeutic responses to acupuncture in patients with PD.

This study is very innovative in acupuncture but the main limitation is the small sample size (n=6).

Minor comments.

  1. In page 1, the 2nd sentence in introduction section “As a complementary and alternative therapy, acupuncture has been widely used to ……”. No need to say “As a complementary and alternative therapy,”. Please delete this.
  2. Considering main limitation of small sample size in this study, it need additional sentences in conclusion in abstract and at the end of discussion section. Something like “Further research is needed to confirm this novel finding”.

Author Response

RE: diagnostics-1368249, Investigation of therapeutic response markers for acupuncture in Parkinson's disease: An exploratory pilot study

Dear Editor and Reviewers,

We appreciate the Reviewers’ helpful comments and suggestions. To address the issues raised, we have performed additional statistical analyses and provided further explanations. The responses to the Reviewers’ comments are presented below.

Response to the Reviewer #1’s Comments

This preliminary pilot study investigated the effect of acupuncture on transcriptome alterations and gene expression response in patients with PD and found that up-regulation of specific gene ANKRD22 and SYN1 and suggests these are potential biomarkers to predict/monitor therapeutic responses to acupuncture in patients with PD. This study is very innovative in acupuncture but the main limitation is the small sample size (n=6).

Minor comments:

  1. In page 1, the 2nd sentence in introduction section “As a complementary and alternative therapy, acupuncture has been widely used to ……”. No need to say “As a complementary and alternative therapy,”. Please delete this.

Response

Thank you for your helpful comment. As suggested, we have removed that sentence in the revised manuscript. (Line 37)

  1. Considering main limitation of small sample size in this study, it needs additional sentences in conclusion in abstract and at the end of discussion section. Something like “Further research is needed to confirm this novel finding”.

Response

As suggested by the reviewer, we have highlighted the need for further research, wherever applicable.
- In the abstract section: “Further research is needed to confirm these findings in a large sample of patients with PD.” (Line 30-31)
- In the discussion section: “Nevertheless, further research is needed to confirm therapeutic response biomarkers to acupuncture in patients with PD.” (Line 372–373)

Reviewer 2 Report

This is a very interesting and innovative paper, with some shortcomings.

My first main concern is that the authors assume a degree of knowledge and understanding about gene expression, sequencing and transcriptomic analysis that many readers who are specialists in other fields may not have. Terms (e.g. ‘fold change’ and ‘density plot’) and abbreviations are introduced without sufficient explanation. A section of more general background information should be added to the Introduction, and a brief glossary of terms and acronyms could be added as an Appendix. At the very least, abbreviations (e.g. alpha-syn, p11) should be spelled out in full when they first occur in the text (not just in the Abstract, as is done for qPCR, for instance).

The plots could also be interpreted more explicitly (e.g. the colours in Figure 3 and Supplementary Figure S3, or the Volcano plot in Figure S2). How is ‘density’ defined in Figure 2A? And is it really feasible to label individual points (or even clusters) in the scatter plot (Figure 2C) so precisely, for such a small sample? Figure 3 is more convincing, overall, but it is not 100% clear to me how the PD biomarker fold change blood and brain results were derived: change with respect to what as the baseline? The little dendrogram (?) in Figure 3B (right) is not explained.

The authors are well aware that over-generalisations should not be made on the basis of results from their small sample of patients, but still hypothesise that “transcriptomic changes in the blood following therapeutic interventions would reflect neural changes in patients with PD” [line 57]. It would perhaps be sensible to use the words ‘could perhaps’ instead of ‘would’ here. Hopefully, in future research using a larger sample, they will be able to replicate some of their present results. This should be one of their explicit aims.

My second main concern is that their evidence for genetic biomarkers in PD derives from a very limited number of published sources. If the involvement of these biomarkers in PD has been replicated by other groups of researchers, this should be stated. If not, this should be made very clear!

I would also like to see some references on verification of results from the various methods mentioned in section 2.6 on qPCR. Simply stating that the manufacturer’s instructions were followed does not guarantee that the results are correct!

In Section 2.7, Statistical Analysis, it could be helpful to the reader to justify use of Welch’s t-test and explain why both Student’s and Welch’s -tests were needed. Use of Pearson’s r rather than another method of correlation analysis suited to very small numbers should also be justified.

Some of the p-values given (e.g. p = 0.0575 for ANKRD22 at V5) can hardly be considered as indicating that ANKRD22 “significantly increased following acupuncture”! Would some method such as Bootstrapping be useful to improve significance?

The Benjamini-Hochberg method was used (caption to Figure S2), but no indication is given on how the false discovery rate was chosen. This should be made clear.

My overriding concern is that the authors mention that the clinical intervention used was acupuncture, but provide NO information whatsoever on what the treatment was or how it was administered. In their earlier study on acupuncture and gait disturbance in PD (Jang et al. 2020) they merely state that “participants in the intervention group received acupuncture 2 days per week for 4 weeks. Acupuncture was performed on the dorsal side. All acupuncture treatments performed in the intervention group complied with the STRICTA (Standards for Reporting Interventions in Clinical Trials of Acupuncture; 2010) guidelines.” This is totally inadequate. If you cite STRICTA, you should at the very least follow the guidelines themselves!!!! In their 2020 paper, the authors do not even state whether manual or electroacupuncture was used. This is critical to an understanding of the effects of treatment. After all, lying down in a comfortable clinic room with a sympathetic practitioner might itself improve hypometric gait. The acupuncture treatment itself may or may not be an important factor in this.  

As no information on treatment was provided in their earlier paper, they MUST provide it here, and follow the STRICTA guidelines themselves when doing so. Otherwise, my opinion  is that this paper should not be published.

There are a number of basic spelling and grammatical mistakes in the Supplementary materials that should be checked (’doulbe’, ‘remaining the five’, etc.), and punctuation mistakes in the main text (e.g. use of ‘;’ rather than ‘,’ in the caption to Figure 1), or use of lower case ‘p’ following a full stop (line 60).

Future directions: The authors used the GAITrite system. In many physiological investigations, complexity and entropy methods have been shown to provide more information than analysis based on traditional measures such as those provided by GAITrite (e.g. Coates et al. 2020). It would perhaps be interesting to add such a method to their analysis in future research.

In conclusion, I very much hope these authors will develop the work they have started in their first two papers on the effects of acupuncture on hypometric gait in PD, but using a larger sample and paying more attention to the treatment administered, other possible factors responsible for patient improvement, and why effect size appears to be greater at V5 than V8.

Reference:

Coates L, Shi J, Rochester L, Del Din S, Pantall A. Entropy of real-world gait in Parkinson's disease determined from wearable sensors as a digital marker of altered ambulatory behavior. Sensors (Basel). 2020 May 5;20(9):2631. doi: 10.3390/s20092631.  

Author Response

RE: diagnostics-1368249, Investigation of therapeutic response markers for acupuncture in Parkinson's disease: An exploratory pilot study

Dear Editor and Reviewers,

We appreciate the Reviewers’ helpful comments and suggestions. To address the issues raised, we have performed additional statistical analyses and provided further explanations. The responses to the Reviewers’ comments are presented below.

Response to the Reviewer #2’s Comments

This is a very interesting and innovative paper, with some shortcomings.

  1. My first main concern is that the authors assume a degree of knowledge and understanding about gene expression, sequencing and transcriptomic analysis that many readers who are specialists in other fields may not have. Terms (e.g. ‘fold change’ and ‘density plot’) and abbreviations are introduced without sufficient explanation. A section of more general background information should be added to the Introduction, and a brief glossary of terms and acronyms could be added as an Appendix. At the very least, abbreviations (e.g. alpha-syn, p11) should be spelled out in full when they first occur in the text (not just in the Abstract, as is done for qPCR, for instance).

Response

  1. Thank you for pointing this out. As per the Reviewer’s comment, we have provided explanations for the following terms:
    - fold change: “A fold change describes the ratio of two values, measuring how much a quantity changes between an original and a subsequent measurement.” (Line 112–113)
    - density plot: “To observe the distribution of expression changes by gene groups, we analyzed the data using a density plot, a smoothed version of histograms.” (Line 207–208)
  2. We have included more general background information about transcriptomic analysis in the Introduction section as follows: “The advent of high-throughput sequencing technology enables genome-wide analysis that can investigate the transcriptome, which is the set of all RNA transcripts [10]. Therefore, several transcriptomic studies have employed genome-wide expression profiling using RNA sequencing (RNA-Seq) data to elucidate the pathogenic mechanisms of PD and to identify genetic biomarkers by examining changes in expression levels induced by disease state or treatment [9].” (Line 46-51)
  3. We have also provided a glossary of abbreviations in the Appendix of Supplementary Information.
  4. We have also provided the definitions for the following abbreviations:
    - α-syn: “alpha-synuclein (α-syn)” (Line 41)
    - p11: “p11, also known as S100 calcium-binding protein A10” (Line 54–55)

  1. The plots could also be interpreted more explicitly (e.g. the colours in Figure 3 and Supplementary Figure S3, or the Volcano plot in Figure S2). How is ‘density’ defined in Figure 2A? And is it really feasible to label individual points (or even clusters) in the scatter plot (Figure 2C) so precisely, for such a small sample? Figure 3 is more convincing, overall, but it is not 100% clear to me how the PD biomarker fold change blood and brain results were derived: change with respect to what as the baseline? The little dendrogram (?) in Figure 3B (right) is not explained.

Response

  1. Thank you for your valuable comment. We have included the explanations in the legends of Figure 3 and Supplementary Figure S3.
    - Figure 3B: “Red and blue colors denote increased and decreased expressions, respectively.” (Line 266)
    - Supplementary Figure S3: “Red and blue colors denote increased and decreased expressions, respectively.” (Supplementary Information)
  2. We have also included a legend for the colors used in Supplementary Figure S2 as shown below.
  3. We have included the definition of density in the legend of Figure 2A: “The density was defined as the proportion of genes with a specific fold change among all genes.” (Line 221–222)
  4. The scatter plot in Figure 2C represents the change in scores for 7079 pathways in GO BP gene set as illustrated in Supplementary Table S5. We had only labeled a few points (pathways) related to synapse transmission. We have revised the legend of Figure 2C as follows: “Each point represents a changed set of scores in pathways of the gene set.” (Line 224–225)
  5. The fold-change values of PD biomarkers in Figure 3A were obtained from the study by Green et al. (2017). The values were derived by calculating the difference in expression levels in patients with PD versus healthy controls. The baseline was the expression level in healthy controls. We have revised the legend of Figure 3A as follow: “Three PD biomarkers consistently decreased in living brain and blood transcriptome versus healthy controls.” (Line 261–262)

  1. The authors are well aware that over-generalisations should not be made on the basis of results from their small sample of patients, but still hypothesise that “transcriptomic changes in the blood following therapeutic interventions would reflect neural changes in patients with PD” [line 57]. It would perhaps be sensible to use the words ‘could perhaps’ instead of ‘would’ here. Hopefully, in future research using a larger sample, they will be able to replicate some of their present results. This should be one of their explicit aims.

Response

  1. Following the reviewer’s comment, we have changed the words as suggested in the revised manuscript. (Line 62)
  2. As suggested by the reviewer, we have included sentences implying the need for future research in the revised manuscript as follows:
    - In the abstract section: “Further research is needed to confirm these findings in a large sample of patients with PD.” (Line 30–31)
    - In the discussion section: “Nevertheless, further research is needed to confirm therapeutic response biomarkers to acupuncture in patients with PD.” (Line 372–373)

  1. My second main concern is that their evidence for genetic biomarkers in PD derives from a very limited number of published sources. If the involvement of these biomarkers in PD has been replicated by other groups of researchers, this should be stated. If not, this should be made very clear!

Response

Thank you for your insightful comment. We aimed to identify possible therapeutic markers from previously reported PD biomarkers replicated by independent studies. Three biomarkers, ANKRD22, IL1R2, and MARCH1, were reported as consistent in both studies on the living brains [Benoit et al. (2020)] and blood [Infante et al. (2016)] of patients with PD. SYN1 was reported as a PD biomarker in four studies with different samples and analysis [Hänzelmann et al. (2013); Miller et al. (2006); Simunovic et al. (2009); Chandrasekaran and Bonchev (2003)].

We agree that theses 6 references are very limited compared to the numerous studies on PD biomarkers accumulated in the past few decades. However, the reported genetic biomarkers were heterogenous between the research groups. Therefore, we considered that the PD biomarkers that were replicated in the few literatures are evident. We have mentioned this point in the revised manuscript as follow: “Among the previously reported PD biomarkers that were replicated in independent studies, we identified genes whose expression changes correlated with the therapeutic response to acupuncture.” (Line 242–243).

  1. I would also like to see some references on verification of results from the various methods mentioned in section 2.6 on qPCR. Simply stating that the manufacturer’s instructions were followed does not guarantee that the results are correct!

Response

Thank you for your helpful comment. Following the reviewer’s comment, we have included a detailed method and cited the appropriate references for cDNA synthesis and qPCR as follows: “Total RNA (500 ng) isolated from whole blood samples was reversed transcribed into cDNA using the Superscript III First-Strand Synthesis System (Invitrogen, Carlsbad, CA, USA). After diluting cDNA to a constant concentration, qPCR was performed using PowerSYBR Green PCR Master Mix (Thermo Fisher Scientific, Waltham, MA, USA). The qPCR reactions were set up in triplicate with a total volume of 20 µL: 2 µL of cDNA template, 1 µL of each primer, 10 µL of Power SYBR Green PCR Master Mix (2´), and 6 µL of nuclease-free water. qPCR was performed using QuantStudio 6 Flex Real-Time PCR System (Applied Biosystems, Foster City, CA, USA) and reaction condition was as follows: 40 cycles of DNA denaturation at 95°C for 15 sec and annealing/extension at 60°C for 1 min. The average cycle threshold (Ct) value of the biomarker genes was calculated post normalization to that of the GAPDH gene and then the fold increase or decrease in gene expression compared to V1 sample was determined [24].” (Line 141–152)

After cDNA synthesis, we checked the quality of cDNAs by conventional PCR for GAPDH gene as below. Please see the picture for reference only.

PCR Mixture: Reaction volume 25 µL (2× PCR Master Mix 12.5 µL, each primer 1 µL, cDNA 1 µL , D.W 9.5 µL), 2× PCR Master Mix: BAT1113, BioAssay Co., Ltd, Republic of Korea)

PCR condition: 1 cycle at 95°C for 2 min, 30 cycle at 95°C for 20 sec/56°C for 30 sec/72°C for 30 sec, and 1 cycle at 72°C for 5 min.

  1. In Section 2.7, Statistical Analysis, it could be helpful to the reader to justify use of Welch’s t-test and explain why both Student’s and Welch’s t­-tests were needed. Use of Pearson’s r rather than another method of correlation analysis suited to very small numbers should also be justified.

Response

We sincerely appreciate the reviewer’s suggestion for statistical analysis. We re-examined the statistical analysis of all results in this study by a statistician. Unfortunately, following the analysis by the statistician, some results have changed and we apologize for our mistake regarding the same. The expression of genes in the UD group were not significantly altered after V8 compared to V1 and the levels of ANKRD22 were significantly increased following acupuncture treatment. Although some of the results were not statistically significant, the tendency observed was consistent. This preliminary study was a pilot study aimed to derive candidate markers for large-scale studies. Therefore, the possibility of identifying a candidate marker remains unchanged. We will hopefully perform a large-scale research in the future using this pilot study as a stepping stone. We have revised the important issue as follows, and the manuscript has also been revised accordingly.

  1. For the data in Figure 2B, equal variability was confirmed from the normality test results. Therefore, the Student's t-tests were more appropriate than Welch's t-tests. Re-analysis using Student's t-tests showed that the increase in gene expression in the UD group was significant at V5 (P = 0.00857 → 0.0004) but not at V8 (P = 0.0435 → 0.0582). Therefore, we changed the descriptions for these results and Figure 2B in the revised manuscript as follow:
    - In the Methods section: “The log2 (fold change) values from the RNA-Seq data were statistically analyzed using a two-tailed Student’s t-test.” (Line 160-161)
    - In the Results section: “Compared to the genes in the DD group, the expression of genes in the UD group increased after both V5 (P = 0. 0004) and V8 (P = 0. 0582) (Figure 2B).” (Line 211-212)
  2. For the data in Figure 4A, comparative analysis of the expression levels of the candidate genes in the same patients at V1, V5, and V8 was more suitable for paired t-tests than for Student’s t-tests. After re-analysis using paired t-tests, the statistical significance was changed for SYN1 (P = 0.0296 → 0.0574 at V5 and 0.0006 → 0.0054 at V8) and ANKRD22 (P = 0.0575 → 0.0910 at V5 and 0.0378 → 0.0678 at V8). Most comparisons were not significant (P > 0.05) except SYN1 at V8. However, the significance level of two genes increased with the number of acupuncture treatments when comparing V8 with V5. As this pilot study aimed to derive candidate markers for subsequent large-scale studies, we still intend to present both genes as candidate markers. Therefore, we changed the descriptions for these results and Figure 4A in the revised manuscript as follow:
    - In the Methods section: “The expression levels measured using qPCR were statistically analyzed using a two-tailed paired t-test.” (Line 161-162)
    - In the Results section: “We found that the levels of SYN1 (P = 0.0574 and 0. 0054) and ANKRD22 (P = 0. 0910 and 0.0678) significantly increased showed an increasing trend following acupuncture treatment (V5 and V8, respectively).” (Line 283–285)
    - In the Discussion section: “Moreover, qPCR validation to determine the changes in gene expression patterns in the remaining five patients with PD, who underwent acupuncture, confirmed the possibility of upregulation in ANKRD22 and SYN1 following acupuncture. The statistical significance of the change in the expression level of SYN1 and ANKRD22 was increased during eight acupuncture treatments (V8) compared to five times (V5), suggesting that these therapeutic response biomarkers may increase with cumulative sessions of acupuncture treatment.” (Line 316–322)
  3. Since the normality test was satisfied for the data in Figure 4B, Pearson’s correlation analysis was applied. As pointed out by the reviewer, Spearman’s correlation analysis is generally more suitable for small sample sizes. We have re-analyzed the data of Figure 5B by Spearman’s correlation analysis as below.

Value

Pearson’s

Spearman’s

SYN1

r

0.676

0.600

P

0.0320

0.0667

ANKRD22

r

0.740

0.685

P

0.0143

Although the correlation coefficients and significance levels were changed following re-analysis, the relationship of positive associations were consistent. Therefore, we have included the following sentence for justification of Pearson’s correlation analysis in the revised manuscript: “After the normality test, we performed Pearson’s correlation analysis between gene expression levels and gait parameters.” (Line 162–163)

Refer to coverletter

  1. Some of the p-values given (e.g. p = 0.0575 for ANKRD22 at V5) can hardly be considered as indicating that ANKRD22 “significantly increased following acupuncture”! Would some method such as Bootstrapping be useful to improve significance?

Response

Thank you for your valuable suggestion. As mentioned in the previous response, the statistical significance was changed for SYN1 (P = 0.0296 → 0.0574 at V5 and 0.0006 → 0.0054 at V8) and ANKRD22 (P = 0.0575 → 0.0910 at V5 and 0.0378 → 0.0678 at V8) after the re-analysis using paired t-tests. Since this is a preliminary pilot study that aimed to identify possible candidate markers. We have discussed the increasing trend of the genes following acupuncture in the revised manuscript.

  1. The Benjamini-Hochberg method was used (caption to Figure S2), but no indication is given on how the false discovery rate was chosen. This should be made clear.

Response

For differentially expressed gene (DEG) analysis, we used the edgeR package in R which provided adjusted P-values after correction with the Benjamini-Hochberg method. We chose the false discovery rate as 0.05 for significance. We have clearly explained this point in the legend of Supplementary Figure S2 as follow: “Adjusted P value (Padj) was obtained using Benjamini-Hochberg multiple testing correction from P-values. For significance, the false discovery rate of 0.05 was chosen.”

  1. My overriding concern is that the authors mention that the clinical intervention used was acupuncture, but provide NO information whatsoever on what the treatment was or how it was administered. In their earlier study on acupuncture and gait disturbance in PD (Jang et al. 2020) they merely state that “participants in the intervention group received acupuncture 2 days per week for 4 weeks. Acupuncture was performed on the dorsal side. All acupuncture treatments performed in the intervention group complied with the STRICTA (Standards for Reporting Interventions in Clinical Trials of Acupuncture; 2010) guidelines.” This is totally inadequate. If you cite STRICTA, you should at the very least follow the guidelines themselves!!!! In their 2020 paper, the authors do not even state whether manual or electroacupuncture was used. This is critical to an understanding of the effects of treatment. After all, lying down in a comfortable clinic room with a sympathetic practitioner might itself improve hypometric gait. The acupuncture treatment itself may or may not be an important factor in this.

Response

Thank you for pointing this out. We apologize for our failure to provide the reference. The details of the interventions under the STRICTA are described in the clinical protocol study [Jang et al. (2018)] that was published in our previous clinical study of acupuncture and gait disorders in PD. We have cited this study in the revised manuscript as follows: “Detailed information about the study design, inclusion criteria, interventions, and out-come measurements can be found in the clinical protocol study. [15]” (Line 90–91)

In addition, the STRICTA information is attached below for better understanding. In this study, manual acupuncture was treated for patients with Parkinson's disease for four weeks (two times per week).

Jang, J.H.; Kim, H.; Jung, I.; Yoo, H. Acupuncture for improving gait disturbance in Parkinson’s disease: a study protocol for a pi-lot randomized controlled trial. Eur J Integr Med 2018, 20,16–21.

  1. As no information on treatment was provided in their earlier paper, they MUST provide it here, and follow the STRICTA guidelines themselves when doing so. Otherwise, my opinion is that this paper should not be published.

Response

Thank you for pointing this out. We apologize for our mistake regarding the same. We have addressed the concerns raised in the previous response. We hope that our response and modification to the manuscript, to reflect the same, are satisfactory.

  1. There are a number of basic spelling and grammatical mistakes in the Supplementary materials that should be checked (’doulbe’, ‘remaining the five’, etc.), and punctuation mistakes in the main text (e.g. use of ‘;’ rather than ‘,’ in the caption to Figure 1), or use of lower case ‘p’ following a full stop (line 60).

Response

Thank you for your helpful comment. We have ensured that all typos throughout the manuscript and the Supplementary materials have been corrected.

  1. Future directions: The authors used the GAITrite system. In many physiological investigations, complexity and entropy methods have been shown to provide more information than analysis based on traditional measures such as those provided by GAITrite (e.g. Coates et al. 2020). It would perhaps be interesting to add such a method.

Reference: Coates L, Shi J, Rochester L, Del Din S, Pantall A. Entropy of real-world gait in Parkinson's disease determined from wearable sensors as a digital marker of altered ambulatory behavior. Sensors (Basel). 2020 May 5;20(9):2631. doi: 10.3390/s20092631.

Response

Thank you for your valuable comment. As suggested, we have discussed entropy methods for evaluating the gait of patients with Parkinson's disease in the revised manuscript as follows: “In addition to the GAITrite system used in this study, entropy methods of evaluating the gait based on complexity theory can provide more information for measuring the therapeutic effect in patients with PD [35].” (Line 380–382)

Round 2

Reviewer 2 Report

Thank you for your careful corrections - in particular for providing a reference for the study protocol used (Jang et al. 2018). It would have been preferable to include this in the first version of the paper submitted, instead of, or as well as, the 2020 paper! 

I am a little concerned that the Spearman's correlations are not significant for the scatter plots in Fig 4B. Which normality test did you use? Was this test suited to small samples? It might be sensible (but not obligatory!) to include a phrase in parentheses in the caption to that Figure: " ... positive correlations (although it should be noted that results using Spearman's rank-order correlation were not significant)." 

Reference numbering is not correct now. 

Otherwise, excellent work and an important contribution.

Author Response

RE: diagnostics-1368249, Investigation of therapeutic response markers for acupuncture in Parkinson's disease: An exploratory pilot study

Dear Editor and Reviewers,

We appreciate the Reviewers’ helpful comments and suggestions. The responses to the Reviewers’ comments are presented below.

Response to the Reviewer #2’s Comments

  1. Thank you for your careful corrections - in particular for providing a reference for the study protocol used (Jang et al. 2018). It would have been preferable to include this in the first version of the paper submitted, instead of, or as well as, the 2020 paper!

Response

Thank you very much for your valuable comment. Following on your opinion, we presented both study protocol paper (Jang et al. 2018) and result paper (Jang et al. 2020) in the manuscript of this paper in round 1.

  1. I am a little concerned that the Spearman's correlations are not significant for the scatter plots in Fig 4B. Which normality test did you use? Was this test suited to small samples? It might be sensible (but not obligatory!) to include a phrase in parentheses in the caption to that Figure: " ... positive correlations (although it should be noted that results using Spearman's rank-order correlation were not significant)."

Response

Following the reviewer's comment, we have added this point in the legend of Figure 4B as follow: "(It should be noted that results using Spearman's rank-order correlation were less significant as P-value of 0.0667 and 0.0289 for SYN1 and ANKRD22, respectively.)". (Line 296–298)

In addition, normality was checked using the Shapiro-Wilk test and the Kolmogorov-Smirnov test.

  1. Reference numbering is not correct now.

Response

We have numbered all the reference correctly.

This manuscript is a resubmission of an earlier submission. The following is a list of the peer review reports and author responses from that submission.